# Biseugenol Exhibited Anti-Inflammatory and Anti-Asthmatic Effects in an Asthma Mouse Model of Mixed-Granulocytic Asthma

**DOI:** 10.3390/molecules25225384

**Published:** 2020-11-18

**Authors:** Vitor Ponci, Rafael C. Silva, Fernanda Paula R. Santana, Simone S. Grecco, Célia Regina M. Fortunato, Maria A. Oliveira, Wothan Tavares-de-Lima, Clarice R. Olivo, Iolanda de Fátima L. Calvo Tibério, Kaio S. Gomes, Carla M. Prado, João Henrique G. Lago

**Affiliations:** 1Departament of Chemistry, Universidade Federal de São Paulo, São Paulo 09972-270, Brazil; vitor.ponci@gmail.com; 2Departament of Biological Sciences, Universidade Federal de São Paulo, São Paulo 09972-270, Brazil; rafael.cossi.silva@gmail.com (R.C.S.); fe.paula.roncon@gmail.com (F.P.R.S.); 3Departament of Biotechnology and Health Innovation, Universidade Anhanguera, São Paulo 05145-200, Brazil; grecco.simone@gmail.com (S.S.G.); celiarmfort@gmail.com (C.R.M.F.); 4Departament of Pharmacology, Institute of Biomedical Sciences, Universidade de São Paulo, São Paulo 05508-000, Brazil; cidora@yahoo.com (M.A.O.); wtavares@usp.br (W.T.-d.-L.); 5School of Medicine, Universidade de São Paulo, São Paulo 01246-000, Brazil; clariceolivo@gmail.com (C.R.O.); iocalvo@uol.com.br (I.d.F.L.C.T.); 6Human and Natural Science Center, Universidade Federal do ABC, São Paulo 09210-580, Brazil; kaiosouza.quimica06@gmail.com; 7Departament of Bioscience, Universidade Federal de São Paulo, São Paulo 11015-020, Brazil

**Keywords:** biseugenol, mixed-granulocytic asthma, airway responsiveness, airway inflammation

## Abstract

In the present work, the anti-inflammatory and antiasthmatic potential of biseugenol, isolated as the main component from *n*-hexane extract from leaves of *Nectandra leucantha* and chemically prepared using oxidative coupling from eugenol, was evaluated in an experimental model of mixed-granulocytic asthma. Initially, in silico studies of biseugenol showed good predictions for drug-likeness, with adherence to Lipinski’s rules of five (RO5), good Absorption, Distribution, Metabolism and Excretion (ADME) properties and no alerts for Pan-Assay Interference Compounds (PAINS), indicating adequate adherence to perform in vivo assays. Biseugenol (20 mg·kg^−1^) was thus administered intraperitoneally (four days of treatment) and resulted in a significant reduction in both eosinophils and neutrophils of bronchoalveolar lavage fluid in ovalbumin-sensitized mice with no statistical difference from dexamethasone (5 mg·kg^−1^). As for lung function parameters, biseugenol (20 mg·kg^−1^) significantly reduced airway and tissue damping in comparison to ovalbumin group, with similar efficacy to positive control dexamethasone. Airway hyperresponsiveness to intravenous methacholine was reduced with biseugenol but was inferior to dexamethasone in higher doses. In conclusion, biseugenol displayed antiasthmatic effects, as observed through the reduction of inflammation and airway hyperresponsiveness, with similar effects to dexamethasone, on mixed-granulocytic ovalbumin-sensitized mice.

## 1. Introduction

Asthma is a chronic inflammatory condition that affects more than 200 million people around the globe, according to the World Health Organization [1,2]. The asthma treatment involves achieving control of symptoms and reduction of exacerbations; however, a portion of patients does not respond to regular asthma control therapy, originating a severe asthma phenotype [3]. Oral corticosteroids are commonly prescribed in severe asthmatic patients, and its chronic use is related to a variety of adverse events [4], highlighting the need for new anti-inflammatory, yet non-immunosuppressive, therapies. There are several endotypes of asthma, including mixed-granulocytic, that has high eosinophil and neutrophil counts, frequently related to a more severe airway obstruction, higher rate of exacerbations and increased health-related costs [5]. For this type of asthma, there is no specific treatment available; thus, a therapy gap exists and must be fulfilled [5].

As previously reported [6], a mixture of different herbal extracts, known as Saiboku-To, has been used in Asia and some regions of Europe for treatment of severe asthma. One of the main active compounds from this extract, magnolol, acts as an inhibitor of T-lymphocyte proliferation, resulting in corticosteroid-sparing effects in severe asthmatic patients [7]. This compound possesses high structural similarity to biseugenol (Figure 1), a neolignan isolated from Brazilian plant *Nectandra leucantha*. This plant has traditionally been used, such as other *Nectandra* species, in South America for the treatment of several diseases in humans, including lung disorders caused by acute and chronic airway inflammation [8]. Based on these pieces of evidence and on the results of ADME using an in silico approach to biseugenol, the evaluation of the antiasthmatic effects of this compound using an experimental model of mixed-granulocytic asthma, focusing on airway hyperresponsiveness and inflammation on bronchoalveolar lavage fluid, was performed

## 2. Results

### 2.1. Chemical Characterization

^1^H and ^13^C NMR (nuclear magnetic resonance) as well as HRESIMS (high resolution electrospray ionization mass spectrometry) spectra (see Appendix A) of natural product were compared with those reported in the literature [9], allowing for the identification of biseugenol (Figure 1). Based on the reduced amounts of natural biseugenol, insufficient to perform the biological assays, the preparation of this compound was performed using oxidative coupling of eugenol (70% yield). NMR and HRESIMS data of obtained compound (100% of purity) were identical of isolated natural product. All biological assays were performed using synthetic biseugenol.

### 2.2. In Silico Studies

The druggability of biseugenol was performed *in silico*, using the SwissADME tool to investigate the physicochemical properties, pharmacokinetics (PK) parameters, and drug-likeness. The obtained results on the bioavailability radar (Figure 2) indicate good adherence of biseugenol to all evaluated parameters (except unsaturation due to the presence of two aromatic rings) but better than those obtained to related compound magnolol.

However, according to the physicochemical properties (Table 1) determined to biseugenol and magnolol, both compounds exhibited high adherence to BigPharma filters (Ghose, Veber, Egan, and Muegge), including no violations to Lipinski’s rules-of-five (RO5). Biseugenol and magnolol displayed also moderate solubility in H_2_O, with log P_o/w_ value of approximately 4.2, an important parameter to deliver good lead compounds [10]. Pharmacokinetics analysis indicated high gastrointestinal absorption, the ability to permeant the blood–brain barrier and no interactions with P-glycoprotein. Furthermore, no alert was evidenced for both compounds for PAINS, indicating that similar to magnolol, biseugenol could be considered to perform in vivo assays in order to evaluate its anti-inflammatory/antiasthma activity.

### 2.3. Selection of Dose Regimen of Biseugenol Treatment

The endpoint to select the optimal dose between three different biseugenol regimens was the reduction of eosinophil and neutrophil counts on bronchoalveolar lavage fluid BALF, as they are known to be key players in mixed granulocytic asthma inflammation [5]. Based on these parameters, biseugenol 20 mg·kg^−1^ for four days (20 mg4d) was the only dose regimen capable of reducing both eosinophil and neutrophil cell count on BALF in a statistically significant way (*p* < 0.05), on ovalbumin-sensitized mice, in comparison to 10 mg·kg^−1^ during four and eight days of treatment (Table 2). Thus, the dose of 20 mg·kg^−1^ for four days was selected for further analysis in lung function parameters and comparison with a positive control (dexamethasone).

### 2.4. Effects of Biseugenol on Airway Hyperresponsiveness (AHR)

With regards to lung function parameters, the OVA-sensitized animals showed higher airway hyperresponsiveness to methacholine, as shown by a significant increase in airway resistance and tissue damping curves (R_aw_ and G_tis_, Figure 3A,D, respectively), maximal response values (Figure 3B,E) and percentage of variation in comparison to baseline values (*p* < 0.001, Figure 3C,F). The response of these animals to methacholine was far superior to those seen in traditional Th2 models frequently run on our research group [11]. There were no changes in elastance (H_tis_) in any of the groups evaluated. The animals treated with biseugenol showed reduced AHR as observed by decreased response to methacholine in R_aw_ (Figure 3A) and G_tis_ (Figure 3D), in doses up to 300 µg·kg^−1^, with a significant reduction in maximal responses (*p* < 0.05, Figure 3B,E). Dexamethasone showed greater efficacy reducing AHR to methacholine even in higher doses; however, the difference between maximal R_aw_ and G_tis_ values were not statistically significant in comparison to biseugenol (Figure 3B,E).

### 2.5. Effects of Biseugenol on Inflammation

After the microscopic cell count of BALF, it was observed that the OVA group presented not only increased total inflammatory cell count, but also elevated levels of macrophages, eosinophils, neutrophils, and lymphocytes in comparison to the control group (SAL) (all *p* < 0.001), as shown in Table 2. Eosinophils and neutrophils were especially higher, mimetizing a mixed-granulocytic asthma endotype, as previously aimed (Figure 4B,C). Biseugenol showed efficacy in reducing pulmonary inflammation, as seen by the reduction in eosinophil (*p* < 0.05, Figure 4C), neutrophil (*p* < 0.001, Figure 4B) and total cell count (*p* < 0.001, Figure 4A), in comparison to the ovalbumin-sensitized model. However, there were no changes in macrophage (Figure 4A) and lymphocyte levels (Figure 4E). Dexamethasone was also effective at suppressing inflammation on BALF, with a statistically significant reduction in all cell count parameters (*p* < 0.001, Figure 4A–E). Both biseugenol and dexamethasone were statistically equivalent on the reduction of eosinophil (*p* = 0.250, Figure 4C) and neutrophil counts (*p* = 0.071, Figure 4B).

Finally, positive correlations were found between AHR and cell count levels on BALF through the linear regression analysis of percentage of variation in R_aw_ compared to baseline and total cell count (r = 0.78; *p* < 0.001), eosinophils (r = 0.81; *p* < 0.001) and neutrophils (r = 0.80; *p* < 0.001) (Figure 5).

## 3. Discussion

Asthma is a heterogeneous disease with different endotypes; therefore, it should be treated in an individualized and personalized way [12]. Among the most common endotypes of asthma, there are Th2-high (typically eosinophilic) and Th2-low (also known as Th1/Th17, typically neutrophilic or paucigranulocytic) [13]. Th2-high asthma is usually responsive to treatment with corticosteroids, as Th2-low or Th1/Th17 asthma has predominance of neutrophil infiltration on airways and can lead to a corticosteroid-resistant severe asthma [12,14]. There is also the asthmatic endotype characterized by a high infiltration of both eosinophils and neutrophils on bronchoalveolar lavage fluid, which is denominated mixed-granulocytic asthma and is usually more severe, with high remodeling and reduced pulmonary capacity [15].

In this work, we aimed to reproduce an experimental murine model of mixed-granulocytic asthma [16,17], utilizing ovalbumin grade V with complete Freund adjuvant (containing inactivated M. tuberculosis), to induce increased inflammation with higher concentrations of both eosinophils and neutrophils in comparison with traditional Th2 allergic models of asthma commonly used by our research group [11]. Patients with mixed-granulocytic asthma are commonly more severe, and a recent publication from the International Severe Asthma Registry indicates that more than 50% of these patients make regular use of oral corticosteroids worldwide, with some countries even reaching 90% of usage, despite their known side effects [4,18]. Especially for the mixed-granulocytic endotype of asthma, there is no targeted therapy available [5], and more research around animal models of this endotype is needed for the better understanding of its physiology and to accelerate research for new non-steroidal anti-inflammatory therapeutic agents that could substitute corticotherapy in these patients, as far as they may not respond well to corticosteroid therapy, leading these patients to unnecessary adverse events without the expected efficacy [14,19]. Even though severe asthmatic patients represent only around five percent of total asthmatic patients, they are responsible for more than eighty percent of asthma healthcare costs, as a result of high hospitalization and emergency visits rates [20].

The treatment with biseugenol was capable of reducing both eosinophil and neutrophil concentrations on BALF, in a statistically significant way, and showed non-inferiority to dexamethasone. Thus, biseugenol had similar clinical effects to dexamethasone, as a potent regulator of eosinophilia and neutrophilia, important modulators of this endotype of asthma. As mentioned before, biseugenol is a structurally related compound of magnolol (Figure 1), which has been reported previously to act as an inhibitor of T-lymphocyte proliferation resulting in corticosteroid-sparing effects in severe asthmatic patients [7]. More recent pieces of evidence showed that magnolol promoted reduced BALF inflammation in ovalbumin-sensitized mice through the reduction in Th2 (IL-4, IL-5, IL-13) and Th17 (IL-6, IL-17A) cytokines and IgE levels, also leading to improvements in lung function and the restoration of bronchial tissue architecture, through the modulation of JAK-STAT and Notch 1 signaling pathways [21]. Biseugenol differs to magnolol only by the presence of additional methoxyl groups in the aromatic rings, suggesting that both compounds could act through similar molecular pathways in order to promote these pharmacological effects.

Polyphenols also present immunomodulatory effects through its antioxidant properties. These substances are well known to inhibit enzymes involved in the production of reactive oxygen species (ROS), like xanthine oxidase and NADPH oxidase (NOX), while upregulating antioxidant enzymes as superoxide dismutase (SOD), catalase and glutathione peroxidase [22]. Oxidative stress caused by increased ROS production induces AHR, mucus secretion, epithelial shedding within respiratory cells, and affects smooth muscle contraction [23].

Biseugenol is a biphenyl compound with anti-inflammatory effects that could be related to its potential antioxidant properties, reducing levels of ROS and promoting reduced lung inflammation and AHR. Another possible mechanism by which biseugenol shows anti-inflammatory effects could be through aryl hydrocarbon receptor binding, a ligand-activated transcription factor that belongs to the basic region-helix-loop-helix (bHLH) superfamily, that is linked to the upregulation of IL-22 [24], IL-25, IL-33, TSLP [25], and ROS production [26] in asthma. Published data on the matter also shows evidence that an aryl hydrocarbon receptor ligand could suppress Th17 response in allergic processes [27]. Further studies shall be carried by our group in order to understand the mechanisms of action involved in the anti-inflammatory effects of biseugenol.

The main pulmonary parameters evaluated on this work were central airway resistance, tissue damping (parameter that reflects the viscoelasticity of lung tissue and possibly the resistance of small airways and is known to increase with better lung functioning) and tissue elastance, which reproduces the tissue ability to return to normal state after an inspiration [28]. In both groups treated with biseugenol and dexamethasone, we observed a reduction in airway hyperresponsiveness. Biseugenol showed similar efficacy to dexamethasone on the reduction of maximal response of central airway and tissue damping (R_aw_ and G_tis_, respectively) in ovalbumin-sensitized animals and significant reduced airway hyperresponsiveness to methacholine in all doses ranging from 30 to 300 µg.kg^−1^, highlighting its potential as a modulator of lung hyperresponsiveness in this model of asthma. On all groups evaluated, there were no significant differences in tissue elastance. This is partially expected because asthmatic models do not suffer relevant changes in lung elastance, as, on the other hand, there are variates on other experimental models with parenchymal tissue destruction (i.e., emphysema models) [29]. In this protocol, we found positive correlations between R_aw_ and BALF inflammatory cell parameters, implicating that a reduced inflammation on BALF leads to a reduction in central airway hyperresponsiveness and better lung function. Reduced lung function, bronchial remodeling and an inflammatory profile with eosinophilia and neutrophilia are the main characteristics of severe mixed granulocytic asthma [14].

The discovery of new therapies that could substitute corticosteroid therapy with similar efficacy and less adverse effects is mandatory. Biseugenol at the dose of 20 mg·kg^−1^ seems promising in that matter as far as it showed efficacy in reducing both AHR and lung inflammation in a mouse model of mixed-granulocytic asthma. Further safety studies must be carried out in order to deeply evaluate its safety profile; however, previous findings suggest that biseugenol did not induce liver histopathologies [30] and these preliminary results highlight biseugenol as a potential substitute of corticosteroid therapy or even as a combination drug to associate with low doses of corticosteroids to decrease the risk of adverse effects commonly associated with its chronic use [19]. A possible limitation of this work is that it is not yet possible to translate either the dose of biseugenol or dexamethasone used in this protocol to mixed-granulocytic asthmatic human patients, as further data regarding pharmacokinetics, toxicology and physiologic parameters are needed to address this matter. In conclusion, our results suggest that biseugenol, obtained from a natural source or chemically prepared from simple procedures and reagents (essential for commercial scale-up), could be used as a potential antiasthmatic drug to either substitute or be used as combination therapy with corticosteroids in order to avoid and spare their new metabolic side effects.

## 4. Materials and Methods

### 4.1. General Experimental Procedures

Sephadex LH-20 (Sigma-Aldrich, St Louis, MO, USA) and silica gel 60 F_254_ (Merck, Darmstadt, Germany) were used for column chromatography (CC) and analytical thin layer chromatography (TLC) separations, respectively. Analytical-grade solvents were used for every chromatographic procedure (Labsynth Ltd., Diadema, SP, Brazil). Eugenol was purchased from Sigma-Aldrich (St Louis, MO, USA) and was used without any further purification. ^1^H and ^13^C NMR spectra were recorded on an Ultrshield 300 Bruker Avance III spectrometer (Billerica, MA, USA) operating at 500 and 125 MHz, respectively, using CDCl_3_ (TediaBrazil, Rio de Janeiro, RJ, Brazil) as solvent and TMS as internal standard. MicroTOF QII Bruker Daltonics (Billerica, MA, USA) spectrometer was used to record mass spectra (positive mode).

### 4.2. Plant Material

*Nectandra leucantha* leaves were collected in the Atlantic Forest area of Cubatão city, São Paulo State, Brazil, in March, 2018. The plant material was identified by Prof. MSc. Euder G. A. Martins. A voucher specimen (EM357) has been deposited in the Herbarium of Institute of Biosciences, University of São Paulo, SP, Brazil.

### 4.3. Extraction and Isolation of Biseugenol

*N. leucantha* leaves (200 g) were dried, powdered and exhaustively extracted with *n*-hexane at room temperature in an automatized system ASE 350 (Thermo Fisher Scientific, Waltham, MA, USA). After evaporation of the solvent at reduced pressure 3.5 g of crude *n*-hexane extract were obtained. Part of this material (3 g) was applied to a silica gel column and eluted with a gradient mixture of EtOAc in *n*-hexane. A total of 85 fractions (10 mL each) were collected and combined into four groups (A to D) after TLC analysis. Fraction C (450 mg) was submitted to CC fractionation over Sephadex LH-20 to afford 78 mg of biseugenol (100% of purity as calculated by high performance liquid chromatography—HPLC).

### 4.4. Preparation of Biseugenol

The preparation of biseugenol involved the dimerization of commercial eugenol through an oxidative coupling reaction, according to the procedure previously reported in the literature [9]. Briefly, from 1.64 g of K_3_[Fe(CN)_6_] and 820 mg of eugenol, 560 mg of biseugenol was obtained (70% yield) after recrystallization using absolute EtOH.

*Dehydrodieugenol*. Amorphous white solid, 100% purity by HPLC. ^1^H NMR (300 MHz, CDCl_3_): δ6.74 (d, *J* = 1.9 Hz, 2H), 6.72 (d, *J* = 1.9 Hz, 2H), 5.98 (m, 2H), 5.08 (m, 4H), 3.92 (s, 6H), 3.35 (d, *J* = 6.7 Hz, 4H). ^13^C NMR (75 MHz, CDCl_3_): δ147.2 (C), 140.9 (C), 137.7 (CH), 131.9 (C), 124.4 (C), 123.1 (CH), 115.7 (CH_2_), 110.7 (CH), 56.1 (CH_3_), 39.9 (CH_2_). HRESIMS (positive mode) *m/z* 327.1598 [M + H]^+^ (calculated for C_20_H_23_O_4_ 327.1596)

### 4.5. In Silico Studies

The in silico studies of biseugenol were performed using the SwissADME platform (http://www.swissadme.ch/) to evaluate pharmacokinetics, drug-likeness and medicinal chemistry parameters [10]. This computational tool analyzes diverse parameters, such as (i) Absorption, Distribution, Metabolism and Excretion (ADME); (ii) physicochemical properties (number of heavy atoms, fraction Csp^3^, number of rotatable bonds, number of H-bond donors, and H-bond acceptors); (iii) lipophilicity (log *p* value), water solubility, pharmacokinetics (gastrointestinal absorption, blood-brain barrier permeant, P-glycoprotein substrates), and skin permeation (log K_p_); (iv) drug-likeness with filters including Lipinski (Pfizer), Ghose (Amgen), Veber (GlaxoSmithKline), Egan (Pharmacia), and Muegge (Bayer); (v) alert for pan-assay interference compounds (PAINS); and vi) synthetic accessibility.

### 4.6. Animals and Ethics Statement

Male BALB/c mice aged 6–8 weeks (22–27 g) were acquired from the Animal Facility of the University of São Paulo, São Paulo, Brazil and were housed under controlled light (12 h light/12 h dark; lights on at 8 am) and temperature conditions (23 ± 1 °C), with free access to water and food. All animal care and experimental procedures were conducted in compliance with the rules of the British Pharmacological Society’s Ethics Committee and of the guidelines of the National Council of Animal Experimentation that regulates animal research according to Brazilian Federal Law (Report no. 111/10/03, 2013). All experimental protocols were approved by the internal ethical committee of both the University of São Paulo (#920/2017) and the Federal University of São Paulo (#3025110417).

### 4.7. Study Design, Immunization and Challenge Protocol

Immunization, challenge and treatment protocol were followed as indicated in Figure 6.

Mice were divided at random into six groups: (a) SAL (submitted to saline protocol and intraperitoneally treated with saline solution); (b) OVA (ovalbumin, submitted to the OVA sensitization and intraperitoneally treated with saline solution); (c) 10 mg8d (submitted to the OVA sensitization and intraperitoneally treated with biseugenol 10 mg·kg^−1^ for eight days); c) 10 mg4d (submitted to the OVA sensitization and intraperitoneally treated with biseugenol 10 mg·kg^−1^ for four days); c) 20 mg4d (submitted to the OVA sensitization and intraperitoneally treated with biseugenol 20 mg·kg^−1^ for four days); (d) DX (submitted to the OVA sensitization and intraperitoneally treated with dexamethasone 5 mg·kg^–1^, for four days) [11,30,31]. Treated animals subjected to OVA sensitization received daily biseugenol i.p., since day 21 (if treated for four days) or since day 17 (if treated for eight days) until the end of protocol. Biseugenol was diluted in DMSO (Sigma-Aldrich, St. Louis, MO, USA) (1:4). OVA animals were immunized using OVA (30.0 µg) (grade V, Sigma-Aldrich, St. Louis, MO, USA) dissolved in 25.0 µL saline solution, in the presence of 75.0 µL of Freund’s Complete Adjuvant (Sigma-Aldrich, St. Louis, MO, USA), administered subcutaneously, on day one of protocol. Challenges were carried on days 21, 22 and 23 after the immunization, with 10 µL intranasal of a 400 µg·mL^−1^ solution containing OVA grade II (Sigma-Aldrich, St. Louis, MO, USA) dissolved in phosphate buffered saline (PBS). Control animals were subjected to saline injections at the same time points as the active control groups. A dose–response analysis was carried out with biseugenol at 10 mg8d, 10 mg4d and 20 mg4d in order to find the most efficacious dose regimen based on inflammatory cell parameters of BALF. The chosen biseugenol dose (20 mg4d) was then compared to the DX group as a positive control.

### 4.8. Evaluation of Respiratory Mechanics

On 24th day, after 30 min of the last dose of biseugenol, dexamethasone or saline treatment, all mice were anaesthetized (120 mg·kg^−1^ ketamine + 12.0 mg·kg^−1^ xilazine i.p.), tracheostomized and connected to a rodent ventilator (FlexiVent; SCIREQ, Montreal, Canada) with the tidal volume at 10 mL·kg^−1^, a respiratory frequency of 150 bpm and 3 cmH_2_O PEEP. The jugular vein was cannulated for later injection of PBS and acetyl-β-methyl-choline chloride (MCh, Sigma-Aldrich, St. Louis, MO, USA). Neuromuscular blockage was made by intraperitoneal injection of pancuronium bromide (1 mg·kg^−1^). Oscillatory lung mechanics was performed by producing flow oscillations at different prime frequencies (from 0.25 to 19.625 Hz) for 3 s. Pressure and flow data were obtained and airway impedance was calculated at each frequency [32]. Airway resistance (R_aw_), tissue viscance (G_tis_) and tissue elastance (Htis) parameters were obtained by applying the constant phase model after intravenous injection of PBS and MCh (30, 100 and 300 µg·kg^-1^). The data used were the mean of the points after the PBS injection and the peak response after the injection of MCh 30, 100 and 300 µg·kg^−1^. Still under anesthesia, animals were exsanguinated by vena cava dissection, and BALF was collected.

### 4.9. Cells Counting in BALF

The collection of BALF was performed by introducing 0.5 mL of sterile saline into the mice lungs via a tracheal cannula and withdrawing the fluid into a test tube on ice. This procedure was repeated three times. The fluid collected was centrifuged at 1000 rpm, for 20 min, at 4 °C, and the cell pellet was re-suspended in 300 µL of a solution containing PBS. Total cells were counted using a Neubauer hemocytometer chamber and an optical microscope with a magnification of 40X. BALF differential cell counts were performed using cytocentrifuge slides at 450 rpm for 6 min (Cytospin 2, Shandon Scientific, Pittsburgh, PA, USA). These slides were stained by differential quick stain (Instant-Prov, New-Prov, Paraná, Brazil), and differential counts of at least 300 cells were made according to standard morphologic criteria.

### 4.10. Correlations

A Spearman rank order correlation analysis was performed in order to evaluate the presence of correlations between BALF cell count and lung mechanics parameters for all animals evaluated that were included in both analyses (*n* = 27). A correlation coefficient of r ≥ 0.7 was considered to be significant.

### 4.11. Statistical Analysis

Normality was evaluated by using the Shapiro–Wilk test, and data were expressed as means SE. The parametric data were analyzed by one-way ANOVA followed by the Student–Newman–Keul’s post hoc test, using Sigma Stat software version 11 (CA). The significance level was adjusted to *p* < 0.05.

## 5. Conclusions

In conclusion, this study demonstrated that biseugenol, a metabolite structurally related to anti-inflammatory/antiasthmatic magnolol but displaying better physicochemical and ADME properties, exhibited an expressive anti-inflammatory activity in an experimental model of mixed-granulocytic ovalbumin-sensitized mice at the dose of 20 mg·kg^−1^, as observed through the reduction of both eosinophil and neutrophil counts on bronchoalveolar lavage fluid and by the decrease in lung airway hyperresponsiveness, with similar efficacy to dexamethasone. Biseugenol was isolated from the leaves of *N. leucantha* and easily prepared using a simple procedure (oxidative coupling) and low-cost reagents (eugenol and potassium ferricyanide) in high purity (100%) and yield (70%), Therefore, our results suggest that biseugenol, obtained from a natural source or chemically prepared from simple procedures and reagents (essential for commercial scale-up), could be considered a candidate for further studies related to asthma treatment to either substitute or be used as combination therapy with corticosteroids in order to avoid and spare their new metabolic side effects.

## Figures and Tables

**Figure 1 molecules-25-05384-f001:**
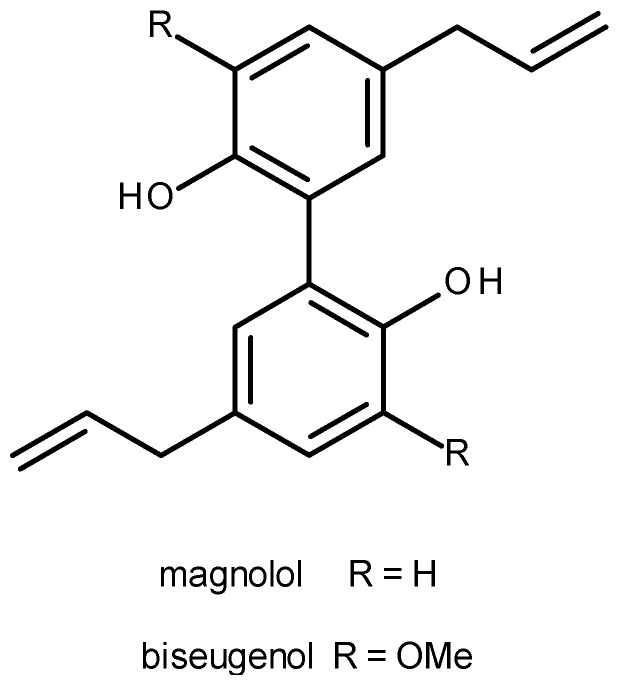
Chemical structures of magnolol and biseugenol.

**Figure 2 molecules-25-05384-f002:**
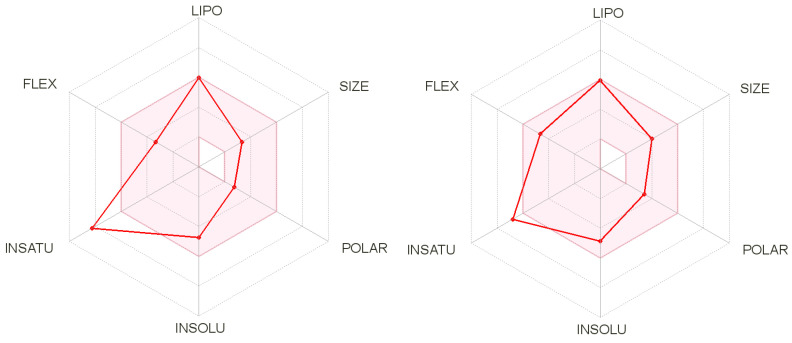
Bioavailability radar for drug-likeness using the SwissADME tool to magnolol (left) and biseugenol (right) demonstrating better adherence of biseugenol to different physicochemical descriptors. The red area represents the optimal range for each property.vb.

**Figure 3 molecules-25-05384-f003:**
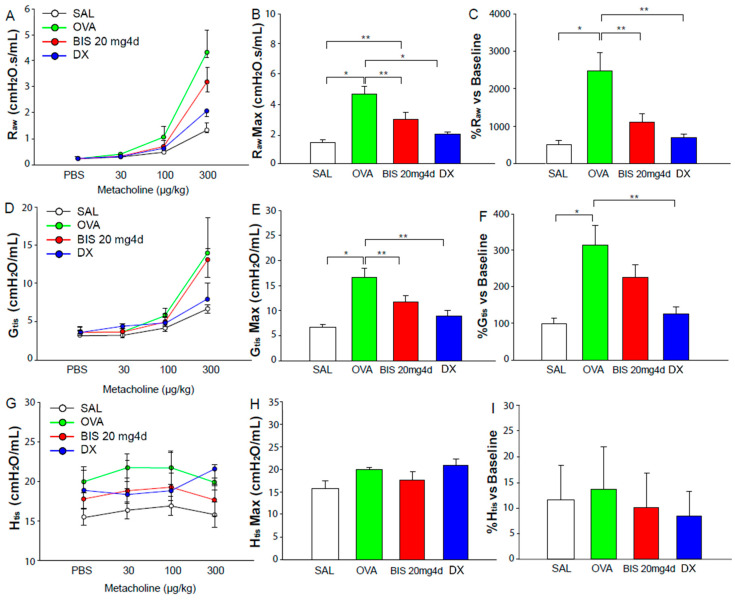
Lung function parameters. (**A**): Dose response curve of R_aw_ (Airway resistance); (**B**): Maximal response of R_aw_; (**C**): % of R_aw_ in relation to baseline values; (**D**): Dose response curve of G_tis_ (tissue resistance); (**E**): Maximal response of G_tis_; (**F**): % of G_tis_ in relation to baseline values; (**G**): Dose response curve of H_tis_ (tissue elastance); (**H**): Maximal response of H_tis_; (**I**): % of H_tis_ in relation to baseline values. SAL (*n* = 8): saline control group; OVA (*n* = 9): ovalbumin-sensitized control group; BIS (*n* = 6): ovalbumin-sensitized group treated with biseugenol 20 mg·kg^−1^ for 4 days; DX (*n* = 7): ovalbumin-sensitized group treated with dexamethasone 5 mg·kg^−1^ i.p. or 4 days. *: *p* < 0.001; **: *p* < 0.05.

**Figure 4 molecules-25-05384-f004:**
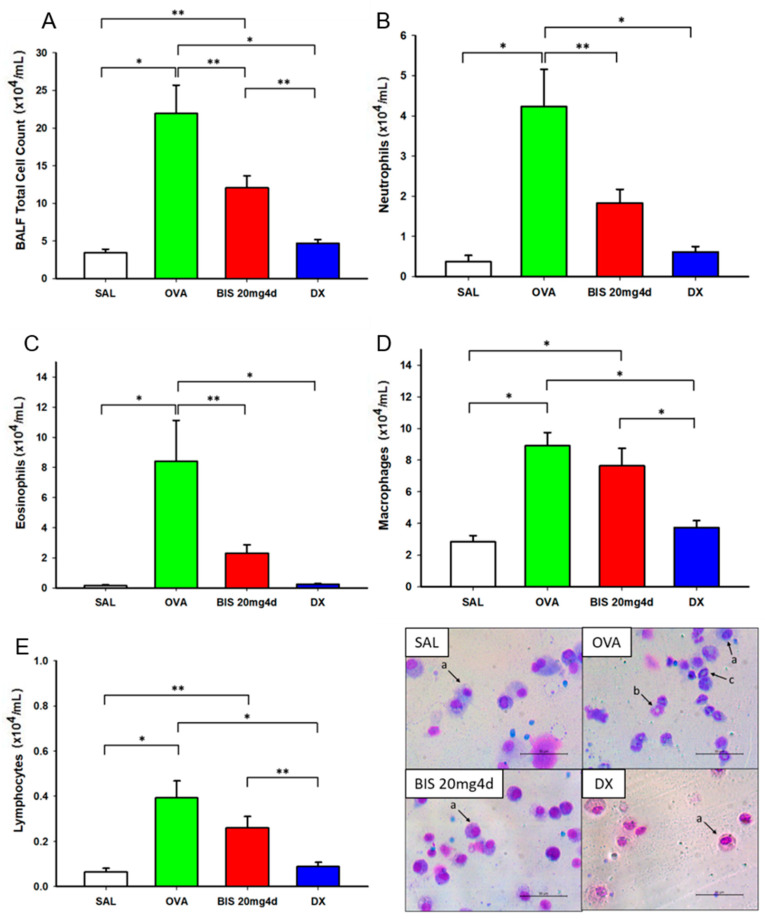
Effects of biseugenol on BALF inflammation and optical microscope images obtained at 40× zoom. (**A**): Total cells; (**B**): Neutrophils; (**C**): Eosinophils; (**D**): Macrophages; (**E**): Lymphocytes counted in BALF. SAL (*n* = 11): Saline control; OVA (*n* = 13): Ovalbumin-sensitized mice treated with placebo; BIS 20 mg4d (*n* = 10): Ovalbumin-sensitized mice treated with biseugenol 20 mg·kg^−1^ i.p. for 4 days; DX (*n* = 9): ovalbumin-sensitized group treated with dexamethasone 5 mg·kg^−1^ i.p. or 4 days. * *p* < 0.001; ** *p* < 0.05.

**Figure 5 molecules-25-05384-f005:**
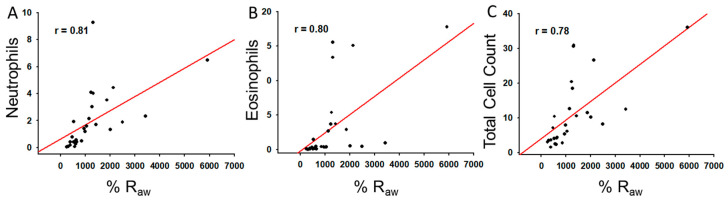
Positive spearman correlations (represented through simple linear regressions graphs) found between %R_aw_ and: (**A**) Eosinophils, (**B**) Neutrophils, (**C**) Total cell count detected in BAL fluid parameters.

**Figure 6 molecules-25-05384-f006:**
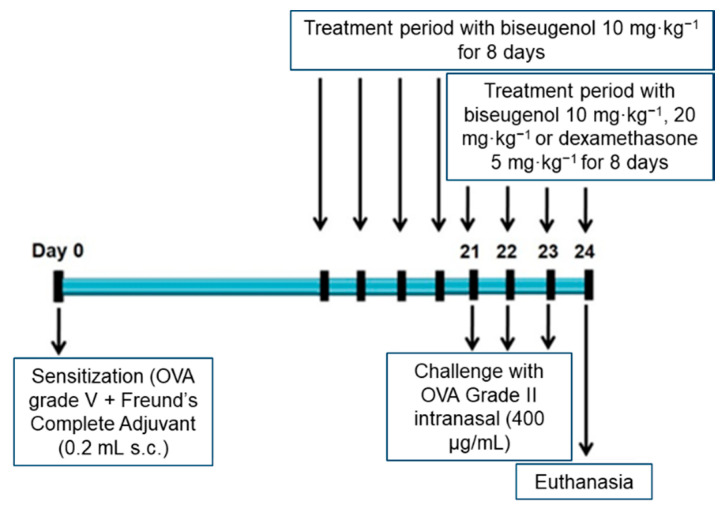
Experimental protocol for the induction of mixed-granulocytic asthma and active treatment period. OVA: ovalbumin.

**Table 1 molecules-25-05384-t001:** Physicochemical properties and ADME predictions [10] for biseugenol and magnolol.

Physicochemical Properties		
	Biseugenol	Magnolol
Number of heavy atoms	24	20
Fraction Csp^3^	0.20	0.11
Number of rotatable bonds	7	5
Number of H-bond acceptors	4	2
Number of H-bond donors	2	2
Lipophilicity		
Log P_o/w_	4.22	4.26
Water Solubility		
Log S	−5.70	−5.47
Solubility	6.5 × 10^−4^ mg·mL^−1^ (2.0 × 10^-6^ mol·L^−1^)	9.1 × 10^−4^ mg·mL^−1^ (3.4 × 10^−6^ mol·L^−1^)
Class	Moderately soluble	Moderately soluble
Pharmacokinetics		
GI absorption	High	High
BBB permeant	Yes	Yes
P-gp substrate	No	No
Log K_p_ (skin permeation)	−4.80 cm·s^−1^	−4.39 cm·s^−1^
Druglikenes		
Lipinski	Yes; 0 violation	Yes; 0 violation
Ghose	Yes	Yes
Veber	Yes	Yes
Egan	Yes	Yes
Muegge	Yes	Yes
Bioavailability Score	0.55	0.55
Medicinal Chemistry		
PAINS	0 alert	0 alert
Synthetic accessibility	3.02	2.49

**Table 2 molecules-25-05384-t002:** Effects of biseugenol on bronchoalveolar lavage fluid (BALF) inflammation.

	Cell Count on BALF (×10^4^)				
	SAL (*n* = 11)	OVA (*n* = 13)	BIS 20 mg4d (*n* = 10)	BIS 10 mg8d (*n* = 10)	BIS 10 mg4d (*n* = 9)
Total cell count	3.43 ± 0.48	21.9 ± 3.76 *	12.0 ± 1.59 **	15.0 ± 3.00	12.4 ± 3.16 **
Macrophages	2.84 ± 0.38	8.90 ± 0.84 *	7.63 ± 1.12	8.48 ± 1.40	5.97 ± 1.01
Neutrophils	0.37 ± 0.16	4.23 ± 0.93 *	1.83 ± 0.34 **	2.07 ± 0.53 **	2.47 ± 0.91
Eosinophils	0.16 ± 0.06	8.41 ± 2.71 *	2.32 ± 0.56 **	4.74 ± 1.40	4.07 ± 1.83 **
Lymphocytes	0.06 ± 0.02	0.39 ± 0.08	0.26 ± 0.05	0.56 ± 0.14	0.41 ± 0.18

SAL: Saline control; OVA: Ovalbumin-sensitized mice treated with placebo; BIS 20 mg4d: Ovalbumin-sensitized mice treated with biseugenol 20 mg·kg^−1^ i.p. for 4 days; BIS 10 mg8d: Ovalbumin-sensitized mice treated with biseugenol 10 mg·kg^−1^ i.p. for 8 days; BIS 10 mg4d: Ovalbumin-sensitized mice treated with biseugenol 10 mg·kg^−1^ i.p. for 8 days. Data are presented as mean SE. * *p* < 0.01 compared with saline control group. ** *p* < 0.05 compared with ovalbumin-sensitized group.

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
