# Peer review of "Biseugenol Exhibited Anti-Inflammatory and Anti-Asthmatic Effects in an Asthma Mouse Model of Mixed-Granulocytic Asthma"

_molecules, 2020, doi:10.3390/molecules25225384_

Round 1

Reviewer 1 Report

The authors present in their research article studies regarding the anti-inflammatory and anti-asthmatic effects of biseugenol. The compound was derived as eugenol from Nectandra leucantha and chemically prepared. Experiments were performed on an experimental mouse model of mixed-granulocytic asthma. The results of the study showed that biseugenol displayed anti-inflammatory properties as observed through the reduction in eosinophils and neutrophils of the bronchoalveolar lavage fluid. Furthermore, biseugenol reduced airway hyperresponsiveness to intravenous methacholine and airway and tissue damping in mice.

Overall, the study presents interesting information adding to the present knowledge regarding the activity of biseugenol. My comments for the authors regarding their research are listed below:

The authors mention that the advantages of biseugenol over magnolol are related to the better physicochemical and ADME  properties of biseugenol. Could the authors include these parameters in Table 1 for magnolol alongside biseugenol.

The authors should provide some additional information regarding the effects of biseugenol on treated mice. The authors mention that no safety concerns were observed in mice treated with biseugenol, could they provide some information on the parameters examined. Furthermore, I suggest changing the statement in the abstract ‘no animals died during the study protocol’ and in the discussion, as it does not  support the use of biseugenol as an anti-asthmatic agent.

Did the authors consider performing histological analysis of lung tissue sections in their study?

The authors should add in the abstract  the concentration of dexamethasone used in the experiments, as they provided for biseugenol. The current information implies that these compounds were used in the same doses. The authors observed similar effects, however, biseugenol was examined  at a 4-fold higher dose than the control drug.

The conclusions of the authors that biseugenol could be used as an anti-asthmatic drug is a bold statement based on the preliminary analysis of the safety of biseugenol. I think it would be more appropriate to state that biseugenol could be considered as a candidate for further studies related to asthma treatment.

The axes descriptions in Figure 3 are not clearly visible.

The manuscript is in most part well-written, however some minor language mistakes are present. Some examples of corrections are listed below:

Line 39: In conclusion

Line 59: acts as an inhibitor

Line 128: lung function parameters

Line 199: and a recent publication

Line 201: known side effects

Reviewer 2 Report

1) line 3, check " effects of on experimental~".

2) line 25, check "work the anti-inflammatory~".

3) which of "P < or p <" is right? For example, there is " P < " in line 113 and " p <" in 131. It needs to be unified.

Author Response

Please, see attachment

Reviewer 3 Report

The manuscript describes a novel action of biseugenol as an anti-inflammatory agent that can be applied in mouse-asthma model. The effects of biseugenol on inflammatory parameters in ovalbumine challenged animals are comparable to that of a steroid (dexamethasone). 

However, in the treatment sheme it is shown that all animals were pre-treated with biseugenol (aryl hydrocarbon receptor inhibitor), before being challenged and repeatedly treated with either biseugenol or dexamethasone. Thus all effects are preventive and not curative. To prove a curative effect the animals should have been treated with the ovalbumine one or two das prior to the compounds. 

1. At least one end-point should be presented with a clear dose-dependent effect of biseugenol.

2. Compared to dexamethsone, biseugenol has a lower efficacy, but to prove this more detailed studies with different dose-ranges are needed.

3. Furthermore, it needs to be discussed if the dose of dexamethasone is comparable to those used for human treatment with other steroids than dexamethasone. This should be better described and discussed. 

Author Response

Please, see attachment 

Round 2

Reviewer 1 Report

The authors have addressed all of my comments and the appropriate modifications have been made in the revised manuscript. 

Author Response

Please see the attachment see the attachment

Reviewer 3 Report

The manuscript has been improved, but the English still needs to be checked:

  1. title has to be improved: "... in an asthma mouse model" or ".. an asthma model.."
  2. line 115: "count" should be either counts or numbers
  3. line 264: "mice" should be "mouse"
  4. line 267: "histopathological alterations" should be "histopathologies" a histopathology presents an alteration of the normal histology.
  5. Table m1: What do the names (Lipinski etc.) indicate? If they are references please provide the link to those.
  6. the statement in the reply: that the disease specific dose of dexamethasone makes it difficult to convert the dose to human is not valid. You are talkinbg about asthma and for this disease there are sufficient publications that did such a conversion of dosage.

Author Response

REVIEWER 3

The manuscript has been improved, but the English still needs to be checked:

Title has to be improved: "... in an asthma mouse model" or ".. an asthma model.."

Attending your suggestion the title was modified to “Biseugenol exhibited anti-inflammatory and anti-asthmatic effects in an asthma mouse model of mixed-granulocytic asthma”.

line 115: "count" should be either counts or numbers

Authors changed “count” by “counts” attending your correction.

line 264: "mice" should be "mouse"

Attending your suggestion, “mice” was changed to “mouse”.

line 267: "histopathological alterations" should be "histopathologies" a histopathology presents an alteration of the normal histology.

Attending your suggestion, “histopathological alterations” was changed to “histopathologies”.

Table m1: What do the names (Lipinski etc.) indicate? If they are references please provide the link to those.

Thanks for your suggestion – as reported in the article (lines 98 – 100) these names refers to filters used in medicinal chemistry as parameters to development of drugs. Specifically, Lipinski's rule of five is a rule of thumb to evaluate druglikeness or determine if a chemical compound with a certain pharmacological or biological activity has chemical properties and physical properties that would make it a likely orally active drug in humans. Attending your suggestion in the title of Table 1 was cited reference 10, which includes discussion concerning these aspects in development of new drugs.

Daina, A.; Michielin, O.; Zoete, V. SwissADME: a free web tool to evaluate pharmacokinetics, drug-likeness and medicinal chemistry friendliness of small molecules. Sci. Rep. 2017, 7, 42717.

The statement in the reply: that the disease specific dose of dexamethasone makes it difficult to convert the dose to human is not valid. You are talking about asthma and for this disease there are sufficient publications that did such a conversion of dosage.

The chosen dose and approach of dexamethasone to treat mice in our protocol were based in several previous publications from our and other groups in the literature [1-6]. As mentioned before, dose translations from animal models to humans are hard to be extrapolated [7], as it must involve physiologic, pharmacokinetic, and toxicology information [8]. Moreover, we must consider that guinea pigs and mice are resistant animals that need higher doses to show the same effect, usually ten times more. If we look into GINA 2020 recommendations for the use of systemic corticosteroids [9], the therapeutic approach is based on daily doses of 50 mg of prednisolone or 200 mg of hydrocortisone in divided doses, 10-14 days. Recommended pediatric doses are usually 0.3 – 0.6 mg/Kg for 5- 7 days. Considering this, we used a dose 10x higher in mice (5 mg/Kg).

[1] Kim S-B, et al. J Ethnopharmacol. 2019, 244: 112083.

[2] Xue K, et al. Int. Immunopharmacol. 2020, 88: 106860.

[3] Niu C, et al. J. Asthma 2019, 56: 11-20.

[4] Toledo AC, et al. Br. J. Pharmacol. 2013, 168: 1736-1749.

[5] Sakoda CPP, et al. Acta Histochem. 2016, 118: 615-624.

[6] Santana F, et al. Biochem. Pharmacol. 2020, 180: 114175.

[7] Reagan-Shaw S, et al. FASEB J. 2008, 22: 659-661.

[8] Blanchard OL, Smoliga JM. FASEB J. 2015, 29: 1629-1634.

[9] Global Initiative for Asthma (GINA). Global Strategy for Asthma Management and Prevention (2020). Available at: https://ginasthma.org/reports/. Accessed: 09/20/2020.